# Integrating Hydrological Connectivity in a Process–Response Framework for Restoration and Monitoring Prioritisation of Floodplain Wetlands in the Ramganga Basin, India

**Manudeo Singh [1,2] and Rajiv Sinha [1,*]**

[1] Department of Earth Sciences, Indian Institute of Technology Kanpur, Kanpur 208016, India
[2] Institute of Geosciences, University of Potsdam, 14476 Potsdam, Germany
[*] Correspondence: rsinha@iitk.ac.in

**Abstract:** Floodplain wetlands are critical for sustaining various ecological and hydrological functions in a riverine environment. Severe anthropogenic alterations and human occupation of floodplains have threatened these wetlands in several parts of the world. A major handicap in designing sustainable restoration and monitoring strategies for these wetlands is the lack of scientific process-based understanding and information on the basin-scale controls of their degradation. Here, we offer a novel approach to integrate the connectivity of the wetlands with the surrounding landscape along with other attributes such as stream density, hydrometeorological parameters, and groundwater dynamics to explain their degradation and then to prioritise them for restoration and monitoring. We hypothesise that the best possible connectivity scenario for the existence of a wetland would be if (a) the wetland has a high connectivity with its upslope area, and (b) the wetland has a low connectivity with its downslope region. The first condition ensures the flow of water into the wetland and the second condition allows longer water residence time in the wetland. Accordingly, we define four connectivity-based wetland health scenarios—good, no impact, bad, and worst. We have implemented the proposed method in 3226 wetlands in the Ramganga Basin in north India. Further, we have applied specific selection criteria, such as distance from the nearest stream and stream density, to prioritise the wetlands for restoration and monitoring. We conclude that the connectivity analysis offers a quick process-based assessment of wetlands' health status and serves as an important criterion to prioritise the wetlands for developing appropriate management strategies.

**Keywords:** floodplain protection; ganga plains; hydrological connectivity; wetland health; wetland management



## 1. Introduction

Wetlands located in the floodplains of a river are integral to their health as they provide essential habitat for aquatic biodiversity, and influence lateral and vertical (groundwater and base-flows) connectivity [1–3]. However, anthropogenic pressure on floodplains and wetlands have led to significant land use changes and encroachment in and around wetlands, pollution from agriculture runoff, sewage and industrial effluents, and lack of ownership to implement mitigation strategies. Globally, there is a growing realisation of the importance of these wetlands for various ecosystem services, including water security for the local population and for maintaining the health of rivers and floodplains. While such anthropogenic stress on floodplain wetlands has been globally recognised [1], its assessment has been difficult [4] primarily because of the lack of process-based understanding of the wetlands and their relationship with the surrounding landscape.

Floodplains are an integral part of a river and a river with a complete floodplain is not just in equilibrium but also in good health. In this context, floodplain wetlands assume a significant importance in river restoration projects. Such restoration efforts would require two fundamental steps: (a) an appropriate wetland classification scheme,

and (b) a structured degradation evaluation approach [5]. The first step could include a hydrologic, geomorphic, biotic classification scheme or a combination of all these. One possible way to do this is to use time-series wetness assessment to classify wetlands based on their hydrodynamics e.g., [6–10] and their geomorphic status e.g., [11–15]. The second step could be implemented at wetland scale or at basin scale. A recent wetland-scale implementation is the use of Wetland Cover Types (WCT) approach that exploits satellite imageries to classify the wetland covers. The WCT approach [16–20] could include, but not limited to, on-screen digitization [21], thresholding of multispectral indices e.g., [17,20], and object-oriented classification e.g., [18,19]. Other approaches for wetland-scale assessment involve the use of various landscape, physical, chemical, biological, and social indicators e.g., [22–24]. However, the WCT and indicator-based approaches are best suitable for in-depth assessment of individual or smaller number of wetlands and may not be appropriate for basin scale mapping of degraded wetlands and their prioritization for restoration. Furthermore, studies in Ganga plains [25–27] and elsewhere [28,29] have shown that regional-scale land use patterns directly impact the freshwater systems such as wetlands. Therefore, for a regional level understanding of processes controlling wetland functions and dynamics, it is imperative to include the land-use/land-cover (LULC) and regional drainage configurations. Hence, a hydrological connectivity-based assessment which accounts for regional scale dynamics [30–32] could be an appropriate step in basin-scale restoration efforts instead of a WCT or indicator-based approach. However, after selecting wetlands best suited for restoration, these WCT and indicator-based approaches can be applied for an in-depth assessment or to gauge the impacts of restoration efforts.

Hydrological connectivity is an emergent hydrogeomorphic property of landscapes [32,33] and results from a complex interaction of anthropogenic (land-use), topographic, biotic, and climatic factors [32,34–36]. In particular, the hydrological connectivity of floodplain wetlands is strongly influenced by the changes in the LULC. For example, the land-cover change from cultivation to grassland in a prairie wetland of North America impeded the snowmelt runoff from the catchment, thereby altering the wetland's hydrology [37]. Similar impacts of land-cover change were observed in large wetlands of the Ganga Basin, such as the Kaabar Tal [27] and the Haiderpur Wetland [25], where rapidly changing LULC resulted in drying and fragmentation of these wetlands. In addition to hydrological connectivity, sediment connectivity also plays an important role in sustaining the wetlands but in a reverse way. High sediment connectivity of the wetland with its catchment results in increased sediment flux [27] and therefore causes siltation which can potentially reduce the 'topographic life' of the wetland [38]. It is therefore imperative that the wetland management and restoration plans include geomorphic (hydrological and sediment) connectivity analysis [36,39], and all efforts should be made to minimise future hydrological connectivity losses [25] and minimise sediment connectivity. There is an urgent need to develop and implement approaches focused on restoring the hydrogeomorphic pathways and associated processes of floodplain wetlands [25,40–42].

We identify the following lacunas in the available approaches for basin-scale wetland health assessment and restoration: (a) most of the methods are site specific and cannot be applied to large number of wetlands, (b) studies including basin-scale wetland health assessments are restricted to wetlands itself and do not consider the land-use/land-cover and hydrological factors operating at regional scale, (c) studies considering wetland-scale as well as regional-scale processes do not account for process dynamics and mostly use a static or instantaneous view, and (d) it should be realised that not all wetlands can be restored due to various logistic and financial constraints, however, there is generally a lack of systematic approach to identify wetlands for their restoration prioritisation based on wetland and catchment scale processes. Therefore, there is a need to develop a protocol that can account for wetland-scale dynamics as well as regional-scale processes and an algorithm to prioritise wetlands for restoration and monitoring.

This work aims to develop a process-response based protocol for basin-scale assessment of wetland health status through hydrological connectivity analysis and their prioriti-

sation for the restoration in the alluvial catchment of the Ramganga Basin (Figure 1). We have integrated hydrological status (stable or degraded) of wetlands for the Ramganga Basin based on our previous work [6] with hydrological connectivity and regional drainage system to generate the priority lists of degraded wetlands that require restoration and of stable wetlands that require monitoring. A novel aspect of this work is that we have separately computed the upslope and downslope hydrological connectivity of the floodplains with the corresponding streams as a function of topographic and LULC factors. We have also analysed time series data of rainfall, LULC, and groundwater to infer the causal factors of degradation of wetlands in a broader sense and have discussed the implications of our work for the restoration of wetlands.

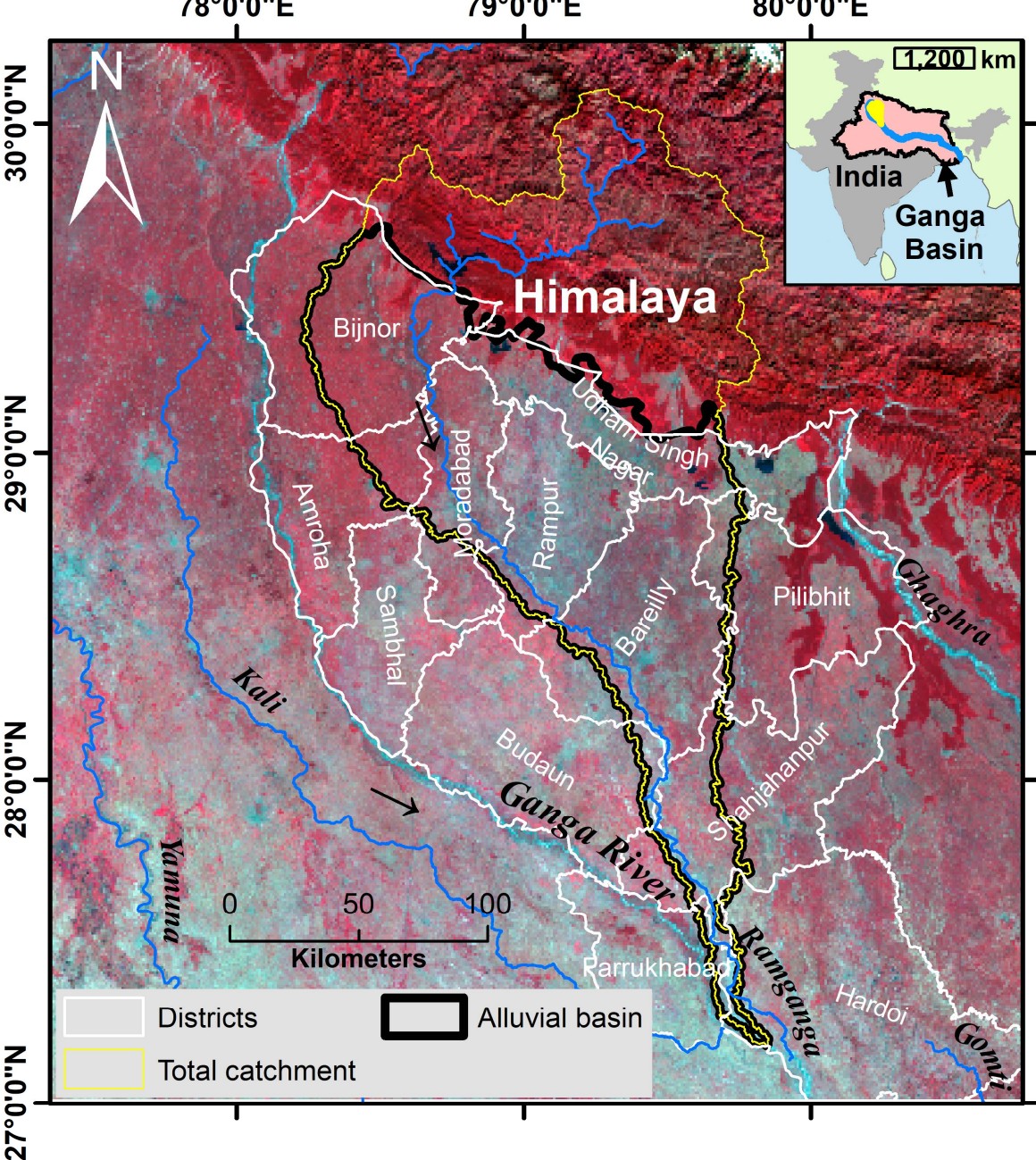

**Figure 1.** The Ramganga floodplains displayed as a standard FCC (Landsat 8, October 2020). The total catchment and alluvial part of the basin have been shown separately. This study pertains to the floodplain wetlands in the alluvial part.

## 2. Study Area and Data Used

The Ramganga basin is drained primarily by the Ramganga river, which is a major tributary of the Ganga River and joins the same near Kannauj district in Uttar Pradesh after traversing through hilly terrains and alluvial plains for about 640 km (Figure 1). The basin is primarily fed by monsoonal rainfall with an average annual precipitation of about 900 mm, while the average daily temperatures vary from 6 °C in winters and 36 °C in summer. More than half of the total area of the basin is composed of agricultural land-use, whereas about 20 percent of the area is covered by grassland, barren land, and urban centres [26]. After emerging from the hilly terrains of the Himalayan foothills, the river drains through densely populated industrial and agricultural lands of the Gangetic plains, viz. Bijnor, Moradabad, Rampur, and Bareilly districts of Uttar Pradesh (Figure 1). The alluvial part of the Ramganga basin is characterised by numerous floodplain wetlands, and our earlier work mapped 3226 wetlands of various sizes, covering a total area of 25,955 ha [6]. One-third of these wetlands lie in Bareilly district alone. Two other prominent districts which host many wetlands are Rampur and Pilibhit. Our previous work showed that almost 70% of the floodplain wetlands in the Ramganga basin are hydrologically diminishing, and ~13% have already been lost [6]. Here we explore the causal factors of degradation of these wetlands using the hydrological connectivity analysis and secondary datasets such as land-use/land-cover, rainfall, and groundwater datasets (Table 1). All datasets used in this study are openly accessible.

**Table 1.** Details of the datasets used in the present study.

| Sl. No. | Dataset | Spatial Details | Temporal Details | Used for | Source |
|---|---|---|---|---|---|
| 1 | Landsat series | 30 m resolution | 1994–2019 (Post-monsoon: Oct-Nov) | NDVI calculation | USGS's Earth Explorer website |
| 2 | CartoDEM | 30 m resolution | DEM for the year 2008 | Topographic factors calculation | Bhuvan website of NRSC |
| 3 | Land-use and Land-cover (LULC) | 60 m resolution | 2005-06 and 2018-19 | LULC changes | Bhuvan website of NRSC (NRSC, 2006; NRSC, 2019) |
| 4 | Wetland extents | Wetland area above 2.25 ha | 1994–2019 | Priority listing | Singh and Sinha (2022a) |
| 5 | Rainfall data-Global Precipitation Measurement (GPM) data | 10 km resolution | Monthly for the period 2002–2019 | Wetland degradation control assessment | Huffman et al. (2019); accessed and analysed using Google Earth Engine |
| 6 | Groundwater data–GRACE Monthly Mass Grids "Equivalent Water Thickness" data | 100 km resolution | Monthly for the period 2002–2017 | Wetland degradation control assessment | Swenson (2012), Landerer and Swenson (2012), Swenson and Wahr (2006). Accessed and analysed using Google Earth Engine |

## 3. Methodology

This work builds upon our previous work on developing a protocol for wetland health assessment in the Ramganga basin based on multi-source and multi-temporal remote sensing data [6]. The novelty of the present work is the integration of connectivity analysis with secondary datasets such as LULC, rainfall data, groundwater dynamics and geomorphic indices (e.g., the proximity of wetlands to river network) to understand and explain the wetland degradation at the basin scale. Based on our analysis, we have also proposed a prioritisation algorithm for restoring and monitoring wetlands in the Ramganga

basin. Figure 2 shows the flow chart of the integrated methodology developed in this work, and the details are provided next.

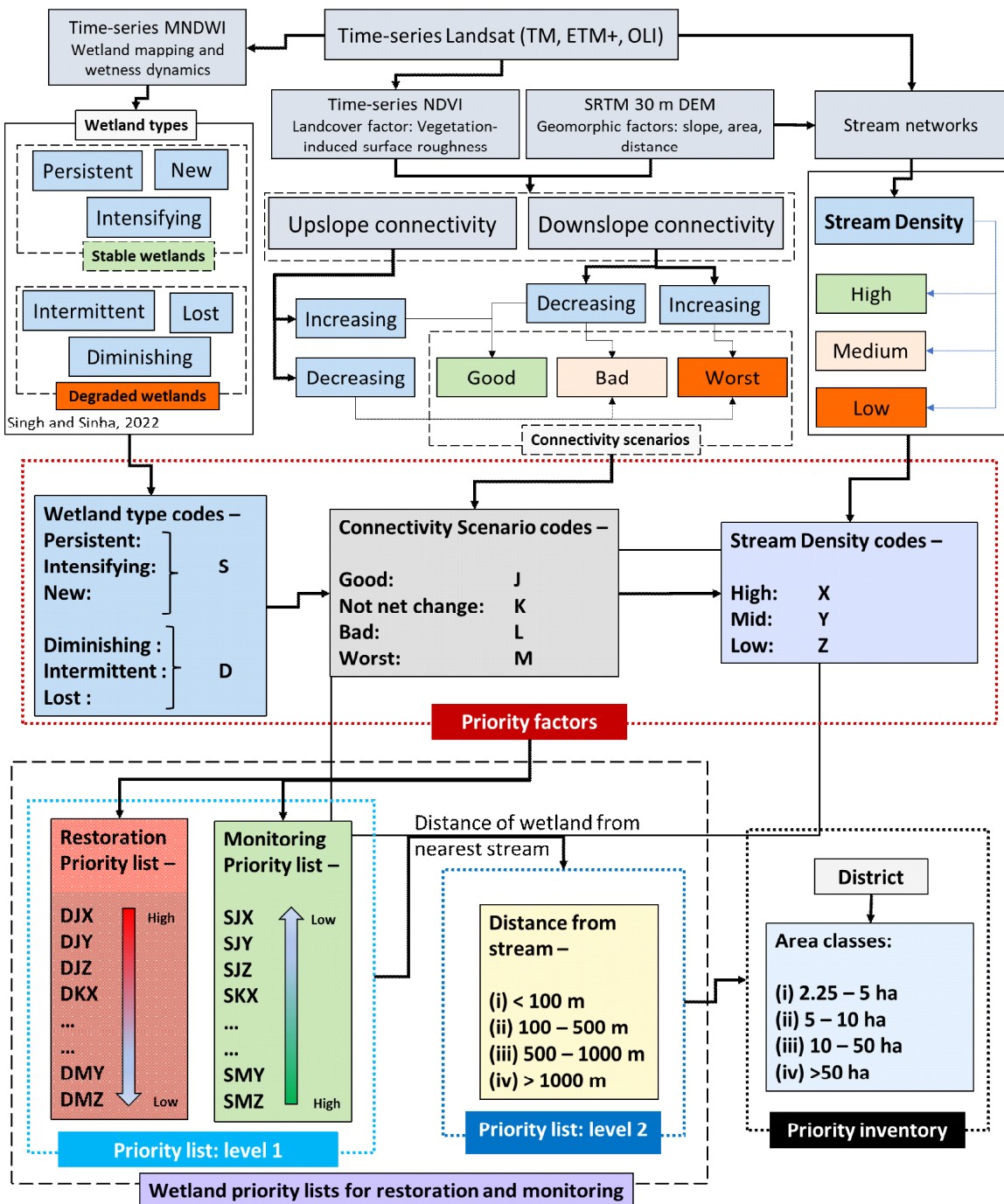

**Figure 2.** Flowchart of the overall methodology followed to generate the priority list of wetlands. Wetland location and their types are taken from Singh and Sinha (2021). Landsat time series period is 1994–2020 for the post-monsoon months (Oct–Nov). MNDWI: modified normalised difference water index; NDVI: normalised difference vegetation index. When both upslope and downslope connectivity increase, the resultant connectivity scenario is 'no net change'—not shown in the flowchart.

### 3.1. Hydrological Connectivity-Based Analysis of Floodplain Wetlands

The hydrological connectivity of wetlands has been assessed to explore its utility for prioritising their restoration and management. At a regional scale, floodplains are composed of heterogenous units, but at a local scale, due to similarity in underlying

factors (LULC, soil-type, stream density, topography), they form homogenous units defined as Connectivity Response Units (CRU) [27,43]. The connectivity values of all features within a given CRU are expected to be the same. Two types of CRUs have been envisaged–spatial [43] and spatio-temporal [27]. The former relates to the structural connectivity of landscapes and provides an instantaneous or static view of the connectivity. The latter accounts for the temporal changes in the landscape and accounts for the dynamics of the connectivity. Therefore, spatial CRUs should be evaluated to assess the connectivity of a landscape at a given time, whereas spatio-temporal CRUs should be evaluated to understand the temporal changes in the landscape and their impact on connectivity.

The hydrogeomorphic characteristics of the wetlands, including their connectivity status, are defined by the attributes of the landscape they are embedded into. Further, the wetlands embedded in such CRUs are expected to exhibit similar connectivity behaviour. Therefore, the estimation of the connectivity dynamics of the host landscape can provide a good assessment of the connectivity status of the wetland. Here, we have used the spatio-temporal CRU approach [27] to estimate the upslope and downslope connectivity status of the landscape as a function of topography and land-cover (Figure 3). We calculated the upslope and downslope connectivity separately since they influence the wetland hydrology differently. For example, if a landscape receives large water inflows from its upslope regions (good upslope connectivity) and has poor outflows (poor downslope connectivity), it provides an ideal scenario for the existence and sustenance of wetlands.

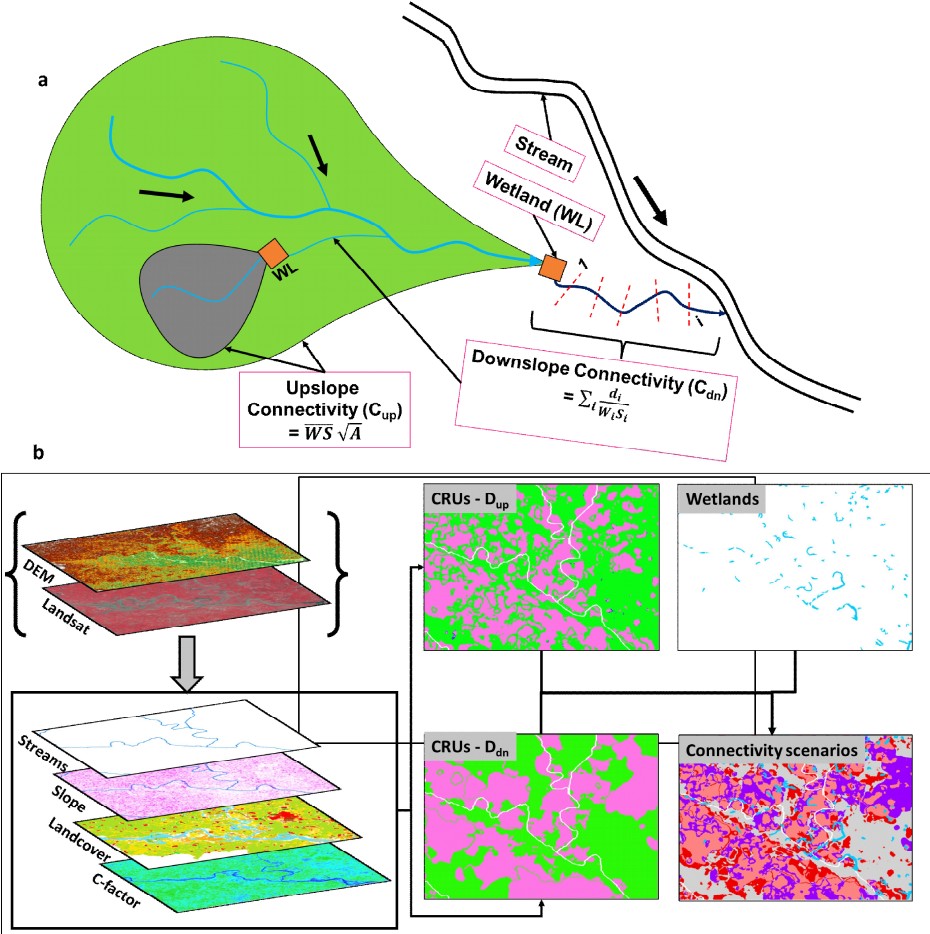

**Figure 3.** (**a**) Wetlands and their nested connectivity scenario from upslope and downslope connectivity response units (CRUs) in an alluvial setting. (**b**) The CRUs are a resultant of underlying properties of a landscape such as topography and landcover which dictate the surface flows.

To obtain the CRUs, we calculated the upslope ($C_{up}$) and downslope ($C_{dn}$) connectivity values at a pixel scale. The upslope connectivity value at a pixel will increase with an increase in (a) upslope area, (b) overall slope of the upslope area, and (c) ease of flow (W). The ease of flow is dictated by surface roughness of the upslope area. Accordingly, $C_{up}$ is directly proportional to upslope area (*A*), upslope slope (*S*), and ease of flow (*W*). Similarly, the downslope connectivity value at a pixel will increase with a decrease in downslope flow-distance (*d*), increase in downslope (*s*), and increase in ease of flow (*w*). Therefore, $C_{dn}$ is inversely proportional to d but directly proportional to *s* and *w*. Similar relationships were used by Borselli, et al. [44] and Cavalli, et al. [45] to calculate a connectivity index (IC) by evaluating upslope and downslope components. Their upslope component is equivalent to the upslope connectivity and their downslope component is an inverse of the downslope connectivity in the present work. These two components of connectivity were calculated using SedInConnect software [46] and the output of downslope component was inversed to get the downslope connectivity values. Accordingly,

$$C_{up} = \overline{W}\,\overline{S}\,\sqrt{A} \tag{1}$$

$$C_{dn} = \sum_i \frac{w_i s_i}{d_i} \tag{2}$$

Here, the upslope factors $A$, $\overline{W}$, $\overline{S}$, are total flow-accumulation area upslope to the given pixel, average weighting factor (a proxy for surface roughness or ease of flow) over A, and average slope over A, respectively. The downslope connectivity is a summation of the factors $d_i$, $w_i$, and $s_i$ which are the horizontal flow-distance from the given pixel to the nearest sink (flow routing path), weighting factor at each pixel in the flow routing path, and the slope at each pixel in the flow routing path, respectively. Further, slope, flow-accumulation area, and horizontal flow-distance are topographic factors and weight (*W*) is a land-cover factor derived from *NDVI* (normalised difference vegetation index) which controls the ease of flow.

CartoDEM (30 m spatial resolution) was used to derive the topographic factor. Since the major land-cover is vegetation, *NDVI* was used to calculate the surface roughness or the ease of flow (weighting factor-*W*). Landsat datasets (30 m) were used to calculate the *NDVI*. It was hypothesised that denser the vegetation, higher is the impedance to the surface flow. Therefore, the barren areas with greater ease of flow would result in higher connectivity. For barren areas, W was given a value of 1 (highest ease of flow) and for densely vegetated regions, W was given a value of 0 (highest resistance to flow). A relationship between *NDVI* and W was developed by sampling the *NDVI* values of high vegetation density regions and barren regions and applying a liner regression between *NDVI* and W. The resulting regression equation is:

$$W = -1.841 \times NDVI + 1.544 \tag{3}$$

The pixel-scale upslope and downslope connectivity were calculated for each year between 1994–2019 for the post-monsoon season. The time series of upslope and downslope connectivity were translated into CRUs by applying the spatio-temporal CRU calculation method of Singh and Sinha [27]. This method uses a combination of Getis–Ord Gi* statistic and the Mann–Kendall trend test to identify the landscape units with similar temporal connectivity behaviour. The Getis–Ord Gi* statistic [47] identifies hot and cold spots in the dataset by comparing the connectivity values of each pixel with the mean values of its neighbouring pixels and with the overall mean values. If a pixel value is closer to the neighbouring pixel's mean value but statistically different than the overall mean value, it is designated as a 'hot spot' (if the pixel's value is higher than overall mean value) or a 'cold spot' (if the pixel's value is higher than overall mean value). In this case, the neighbourhood was defined as 100 m, corresponding to 1 hectare of area around any given pixel. The Mann–Kendall trend test [48,49] was used to establish the statistically significant increasing or decreasing trends of hot and cold spots over time. Since these hot and cold spots define the CRUs, three types of spatio-temporal CRUs were defined—increasing,

decreasing, and no change over the studied time-period (1994–2019). We have used these three CRU types and corresponding wetland sustenance situations to define four different connectivity scenarios—good (J), no net change (K), bad (L), and worst (M) (Table 2). A good scenario for wetland sustenance is defined by a good upslope connectivity (increasing CRU) and a bad downslope connectivity (decreasing CRU) so that the wetland can receive a good amount of water from its upslope region and the received water resides there for an extended period. Similarly, a worst scenario would be when the upslope connectivity is decreasing and the downslope connectivity is increasing, resulting into lower inflows to the wetland from upslope region and a lower residence time of water which would readily move to the downslope region.

**Table 2.** List of criteria used in the prioritisation algorithm.

| Criteria: Level 1 | Criteria: Level 2 | Unique Keys | Level 2: Definition |
|---|---|---|---|
| Wetlandtype | Degraded | D | Lost, diminishing, and intermittent types of wetlands represent hydrologically degraded status and are included in category 'D' |
| | Stable | S | New, intensifying, and persistent types of wetlands represent hydrologically stable status and are included in category 'S' |
| Connectivity Scenario | Good | J | Scenario when upslope connectivity is increasing or registers no change for studied period and downslope connectivity is decreasing |
| | No net change | K | Scenario when no change has been observed either in the upslope or downslope connectivity for the studied period |
| | Bad | L | Scenario when upslope connectivity has not changed with time, but downslope connectivity has increased; or upslope connectivity is decreasing, and downslope connectivity is either not changing or decreasing for studied period |
| | Worst | M | Scenario when upslope connectivity is decreasing, and downslope connectivity is increasing |
| Stream Density (SD) | High SD | X | $SD > 0.21 \text{ km}/\text{km}^2$ |
| | Mid SD | Y | $0.07 \text{ km}/\text{km}^2 > SD < 0.21 \text{ km}/\text{km}^2$ |
| | Low SD | Z | $SD < 0.07 \text{ km}/\text{km}^2$ |

### 3.2. Prioritisation Algorithm: Criteria Used and Justification

We have used a multi-criteria decision approach for the prioritisation of the wetlands which involved three datasets—wetland types, connectivity scenarios, and stream density. In our earlier work, we developed six classes of wetlands based on their physical attributes derived from the wetness index dataset: persistent, intensifying, intermittent, diminishing, new, and lost [6]. In the present work, we combined these classes into two major groups—degraded and stable—to facilitate the prioritisation list (see Figure 2 and Table 2). For degraded wetlands, we have prepared the restoration priority list, and for stable wetlands, we have prepared the monitoring priority list.

Further, our connectivity analysis suggested that the loss of surface hydrological connectivity plays an important role in wetland degradation in this region. Therefore, we have used the connectivity scenarios such as good (J), no net change (K), bad (L), and worst (M) (see Figure 2 and Table 2) as important inputs for developing the priority list.

In addition, we generated the stream density maps with a buffer of 5 km for the basin from DEM-derived flow pathways, and this was also used to prioritise the wetlands. Generally, the areas of higher stream density were given higher priority as they are likely to help in restoring the hydrological connectivity between the channel and wetland (Figure 2).

These three datasets, namely, wetland type, connectivity scenario, and stream density, represent Level 1 criteria of prioritisation. We have considered that restoring connectivity will play the most important role in rejuvenating degraded wetlands. Therefore, we have given the connectivity scenario a higher weightage in prioritisation algorithm compared to stream density. All three Level 1 criteria are further subdivided into Level 2 criteria (Figure 2). Unique alphabetic keys have been assigned to the Level 2 criteria in an ordered fashion. For example, connectivity scenarios have been assigned keys: J, K, L, M in order of increasing influence on wetland degradation. Accordingly, J represents the good connectivity scenario, and M represents the worst. Therefore, a combination of Level 1 criteria results in unique codes (Figure 2). For example, in the case of a degraded wetland (D), which lies in CRUs with a good connectivity scenario (J) and in a region with high stream density (X), the combined unique priority code would be DJX. Accordingly, there are 12 unique priority codes for degraded wetlands and 12 for stable wetlands. The priority codes also identify the primary stressors for individual wetlands. For example, for a wetland with DJX, a wetland-scale stressor would be responsible for its degradation since regional-scale factors, i.e., connectivity and drainage density are in good condition.

The Level 2 keys are alphabetically ordered from best to worst favourable conditions for the wetlands. Therefore, in the case of degraded wetlands, an alphabetically sorted 12 unique priority codes will represent the ease with which a wetland can be restored (Figure 2). For example, a wetland with priority code DJX has more favourable conditions than a wetland with priority code DJY or DKX. Similarly, in the case of stable wetlands, the wetlands with priority code SJX are less vulnerable than those with priority codes SJY, SJZ, etc. Therefore, in the case of stable wetlands, the wetlands with priority codes placed down in an alphabetically sorted list are more vulnerable than those which are placed higher (Figure 2).

Hence, in the case of degraded wetlands, restoration should be prioritised for those wetlands which come first in the alphabetically sorted priority code list. Similarly, in the case of stable wetlands, monitoring should be prioritised for those wetlands which come later in the alphabetically sorted priority code list (or, equivalently, a high priority of monitoring to those which come first in the alphabetically reverse-sorted priority code list). In addition, the distance from the stream and the size of the wetland can also be used to further filter the list of prioritised wetlands (Figure 2).

### 3.3. Hydrometeorological Data and LULC Analysis

We have analysed several secondary datasets to understand the controls of wetland degradation vis à vis connectivity. First, we have analysed basin-scale precipitation data from monthly GPM (global precipitation measurement) data [50] for the period 2001–2022 to create a time series and also to calculate rainfall gain/loss (mm/y) in different parts of the basin in the Google Earth engine (GEE) environment. Second, we computed groundwater deviation trends for the Ramganga basin using the GRACE dataset [51–53]. We also used the GEE environment to assess and evaluate GRACE Lands Mass grid data. This data represents the deviation of equivalent liquid water thickness (cm) from a time-mean baseline calculated for the period 2004–2010 and is a proxy for groundwater dynamics. We evaluated the rate of change in this deviation for the inter-monsoon months of Oct–May.

Finally, we used the LULC data provided by the National Remote Sensing Centre (NRSC) for the time periods 2005-06 [54] and 2018-19 [55] for the Ramganga basin to extract

the major changes between 2005 and 2019 and to relate these with groundwater loss and then to wetland degradation.

## 4. Results

### 4.1. Connectivity of Floodplain Wetlands

Hydrological connectivity of the Ramganga basin was calculated for the post-monsoon season of the years 1994, 1996, 1998, 2002, 2009, 2011, 2014, 2015, 2016, 2017, 2018, and 2019. The wetlands residing within any given CRU inherit their connectivity structure from that CRU. Since the connectivity is calculated with respect to the streams of the basins, the resultant CRUs with the given connectivity status also reflect the connectivity status of the wetlands with the upslope region and downslope pathway. This section discusses the connectivity results of the Ramganga basin and the relationship of surface hydrological connectivity with wetland types. The wetland types are segregated into two classes—degraded (lost, intermittent, diminishing) and stable (new, persistent, intensifying).

The connectivity dynamics, represented by upslope and downslope CRUs, of the alluvial part of the Ramganga Basin with respect to the channels is presented in Figure 4. The upslope and downslope connectivity have three broad trends—increasing, decreasing, and no change. Figure 4a,b show the upslope and downslope components of connectivity, respectively for the Ramganga basin. The centroids of all wetlands and their types (stable/degraded) are also plotted in this figure to understand the relationship of wetland distribution with connectivity. In general, most of the NW regions of the Ramganga Basin (Bijnor and northern Moradabad districts) are exhibiting a decreasing trend in upslope connectivity. The NE region comprising the Bareilly district also shows a decreasing trend in the upslope connectivity (Figure 4a). Most of the central and southern regions of the catchment are exhibiting an increasing trend in upslope connectivity. The downslope connectivity, in general, is showing an increasing trend close to channels and a decreasing trend elsewhere (Figure 4b). Figure 4c shows the resultant of the upslope and downslope connectivity trends for the Ramganga basin, which are classified into four different scenarios, as discussed earlier (Figure 2). The NW regions of the basin are mostly classified into worst and bad scenarios of connectivity. The regions with good scenarios are mostly present in the central and lower parts of the basin.

Based on the distribution of the wetland types within the regions of connectivity scenarios, we observe two broad patterns: (a) direct relationship between wetland types and connectivity scenarios, and (b) inverse relationship between them. In the first case, the degraded wetlands fall within bad and worst connectivity regions and stable wetlands in good connectivity regions. We infer that in such cases, surface hydrological connectivity might be the primary control to sustain the wetlands. In the second case, where we do not observe a direct relationship between connectivity scenarios and wetland types, we infer that surface hydrological connectivity might not be influencing the wetlands as strongly as the other factors, such as vertical connectivity and LULC changes (discussed later). Further, it was observed that the regions of the Ramganga Basin with bad and worst scenarios were in many places devoid of wetlands (e.g., Bijnor and northern Moradabad districts; see Supplementary Materials for district-wise maps and statistics).

We note that Bareilly and Pilibhit districts have the highest density of wetlands among all other districts of the Ramganga Basin, and most of the regions of these two districts are falling under bad and worst scenarios. Additionally, most of the wetlands in these two districts are of diminishing type [6]. It is therefore concluded that the loss of connectivity has played a significant role in the degradation of wetlands in these districts.

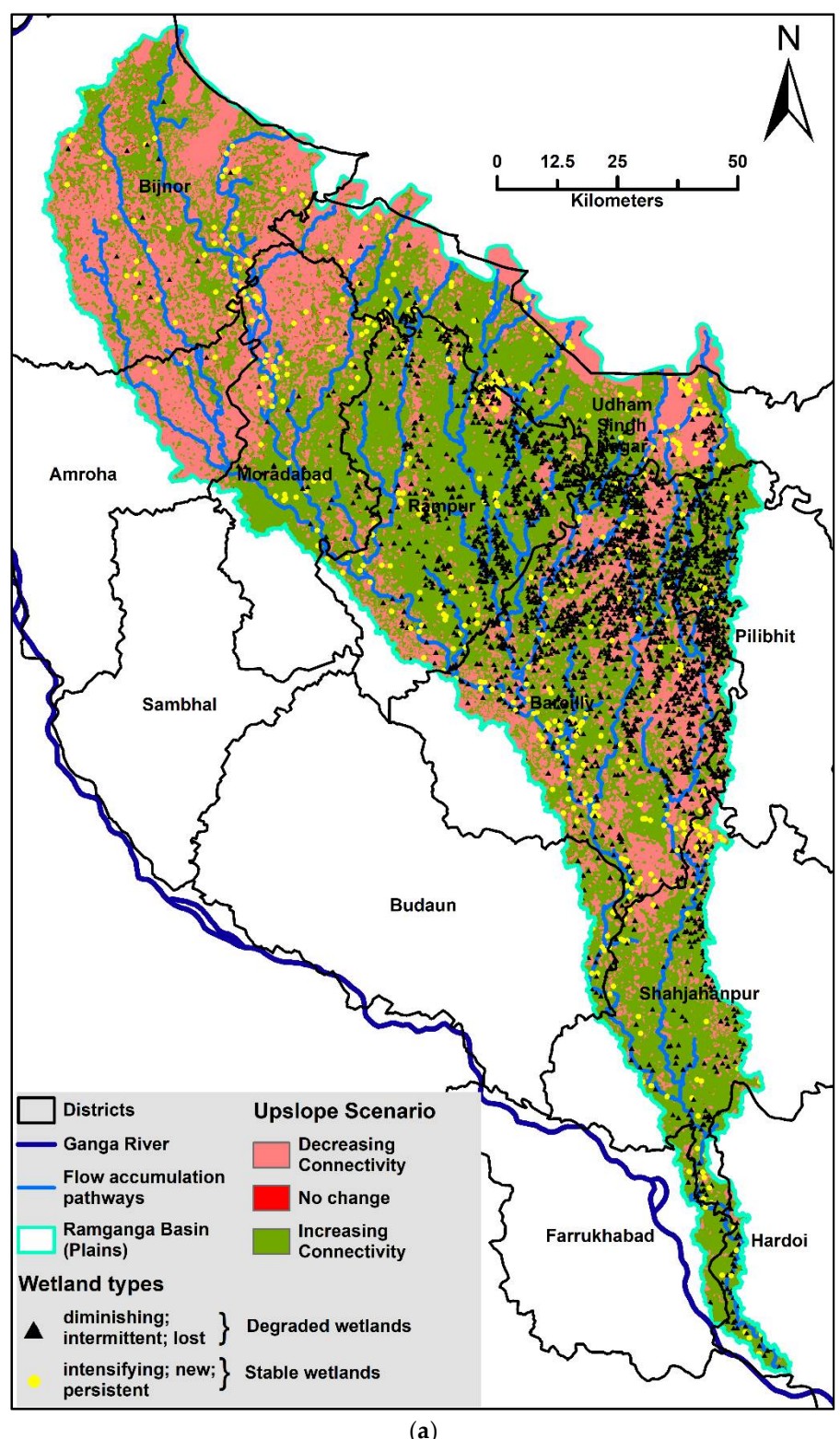

(**a**)

**Figure 4.** *Cont.*

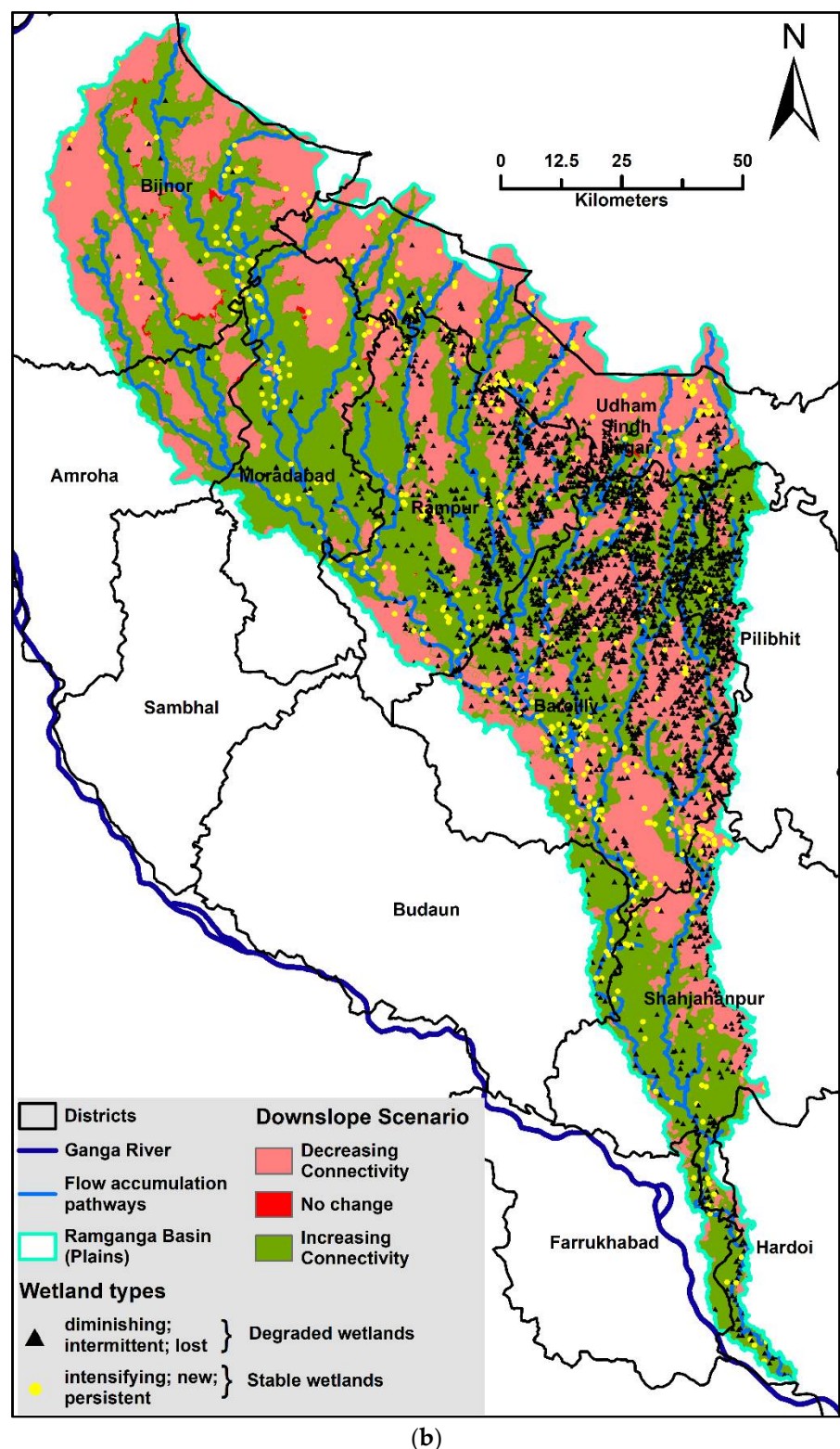

**Figure 4.** *Cont.*

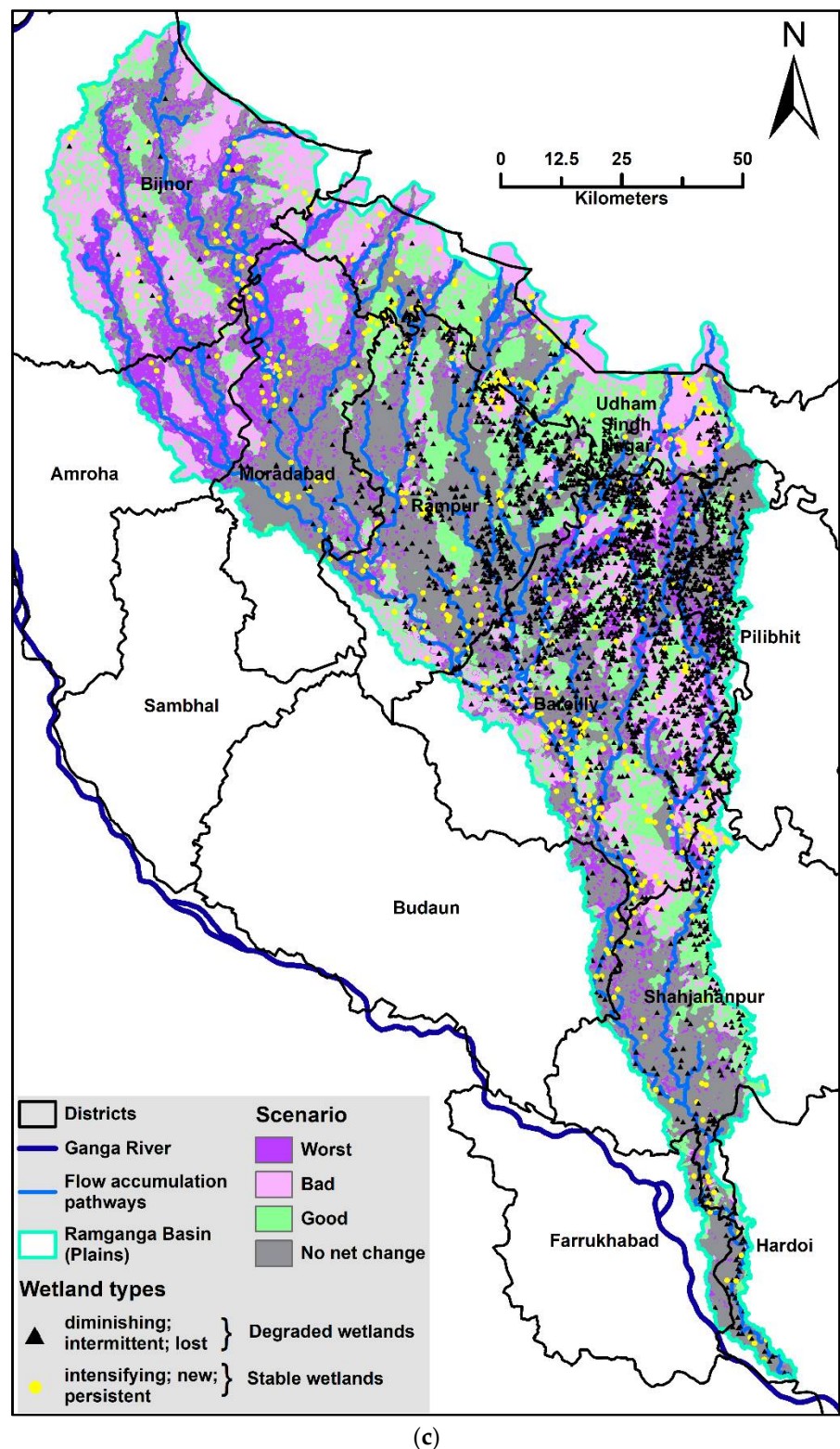

(**c**)

**Figure 4.** (**a**). Overall upslope connectivity trend. The centroids of floodplain wetlands are also plotted. (**b**). Overall downslope connectivity trend between 1994–2019 represented by increasing and decreasing CRU types. The centroids of floodplain wetlands are also plotted. (**c**). Overall scenario of wetland existence as a result of upslope (Figure 4a) and downslope (Figure 4b) connectivity trend between 1994–2019.

Further, Moradabad and Amroha are the districts with regions of predominately worst and bad scenarios of connectivity. There are very few wetlands in Moradabad and Amroha districts, possibly because of the prevailing unfavourable conditions for the sustenance of wetlands. Rampur has a sizable number of wetlands, and most of this district shows a 'no net change' connectivity scenario. However, an interesting result is that several 'diminishing' wetlands in these three districts fall in the good scenario of surface connectivity, suggesting that other factors (e.g., loss of vertical connectivity or other localised anthropogenic factors) might have played a role in the degradation of these wetlands. Bijnor district mostly consists of worst and bad types of scenarios, and there are negligible wetlands in this district, implying the effect of upslope and downslope connectivity scenarios on the existence of the wetlands discussed in Table 2. In Udham Singh Nagar district, several 'diminishing' wetlands are present in the regions exhibiting good scenarios and this implies that other factors besides the surface hydrological connectivity are controlling the wetland hydrodynamics in such regions.

### 4.2. Priority List of Wetlands

Based on the criteria and algorithm discussed above, we have produced a list of prioritised wetlands for each district. A total of 3226 wetlands are present in the Ramganga Basin, out of which 2731 wetlands need restoration and 495 need monitoring. In the restoration category, Bareilly tops the list with 1332 wetlands to be restored, followed by Rampur (360 wetlands). In the monitoring category, again, Bareilly tops the list with 131 wetlands, followed by 93 wetlands in Udham Singh Nagar and 75 wetlands in Rampur. In terms of area, a total of 34,884 ha of wetland are present in the Ramganga Basin, out of which a total of 23,982 ha needs restoration, and 10,902 ha needs monitoring. Area-wise, Bareilly again tops the list for restoration list (10,968 ha), followed by Pilibhit (3931.5 ha), which in turn is closely followed by Shahjahanpur (3899.8 ha) and Rampur (3820.7 ha). In the monitoring list, Udham Singh Nagar surpasses all other districts by a large margin, where an area of 7587.5 ha of wetlands needs monitoring. The second largest area is 960.9 ha for the Bareilly district. The existence of large water reservoirs in the Udham Singh Nagar is the reason for this observation. For district-wise priority listing, see Supplementary Materials.

The distribution of wetlands based on their priority is plotted with the stream location and stream density in Figure 5. The distribution shows that most stable wetlands are situated in high stream density regions. Further, the stable wetlands requiring the least monitoring are mostly situated in the highest stream density regions, indicating a strong influence of streams on their hydrological sustenance. On the contrary, most of the wetlands requiring restoration are situated in the regions of the least stream density and are mostly clustered in the middle of the interfluves. We, therefore, argue that the stream density and distance of wetlands from active streams are important geomorphic controls on floodplain wetlands which in turn influence the hydrological connectivity, thereby creating a complex process–response system (discussed later).

### 4.3. Hydrometeorological Trends and LULC Changes

Time series data of total monthly rainfall and groundwater for the period 2001–2017 (Figure 6a) provide important insights into the hydrometeorological conditions in the Ramganga Basin. Rainfall data show a typical monsoonal pattern with peaks varying between 80 and 100 mm/h throughout the study period and do not show any significant increasing or decreasing trend. However, groundwater level data show a distinct decreasing trend (with respect to the mean of 2004–2010, henceforth called baseline data) between 2001 and 2017 which matches with the increase in crop area until 2014 at the basin scale (Figure 6b). In the post-2014 period, the monsoon crop area kept increasing, manifesting in a sharp decrease in the post-monsoon groundwater level. However, the other crops show a decreasing trend, possibly because of the low availability of groundwater in the non-monsoon period. Between 2002–2005, the average groundwater level deviation from the baseline was about (+)29 cm, which was reduced to (−)47 cm between 2006–2010 and

(−)83 cm between 2011–2016. Therefore, the groundwater level has been severely declining with time with respect to the baseline.

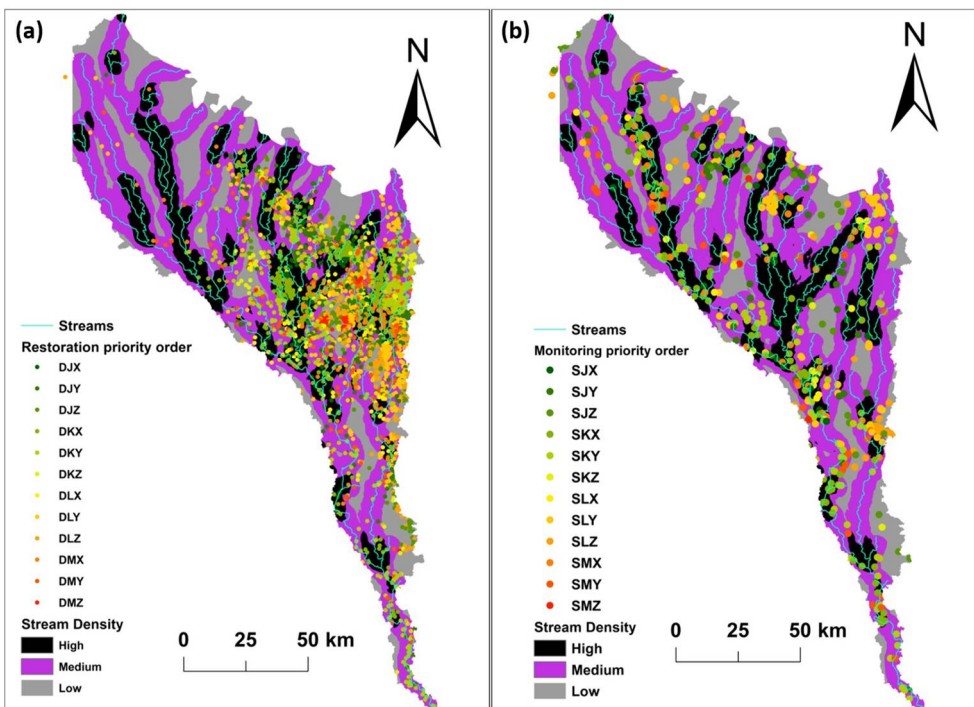

**Figure 5.** Distribution of wetlands based on the priority lists in relation to streams and stream density for (**a**) restoration and (**b**) monitoring lists. Most of the stable wetlands are in the proximity to streams whereas most of the degrading wetlands are in interfluves.

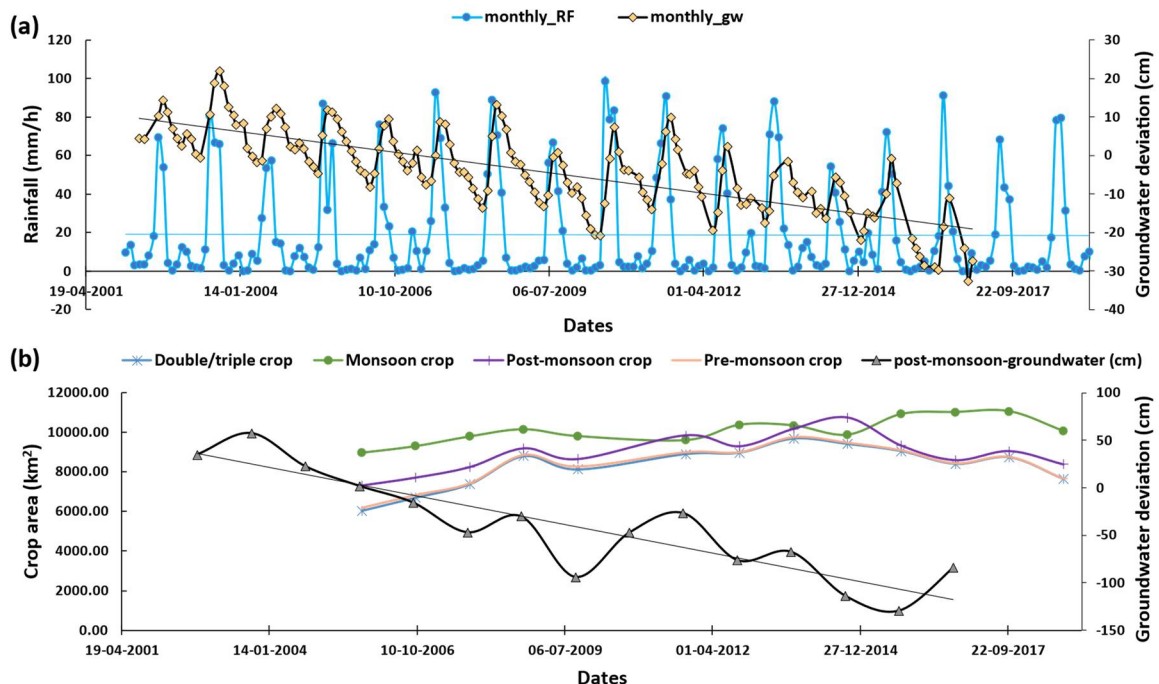

**Figure 6.** (**a**) Catchment-scale monthly trends of groundwater (GW) and rainfall (RF). The lag between RF and GW is evident. Groundwater values are GRACE-derived and represent the monthly deviation of groundwater from a 2004–2010 time-mean baseline. The overall rainfall trends have not changed much. (**b**) Crop areas and post-monsoon groundwater deviation trends at annual scale. Crop areas are steadily rising whereas groundwater is sharply declining over the years.

Figure 7 further illustrates the dynamics of hydrometeorological parameters and LULC vis à vis wetland conditions and connectivity scenarios. While the monthly rainfall data did not show any significant trend, the rainfall loss map (Figure 7a) for the monsoon season shows that several parts of the Ramganga basin fall in the region of rainfall loss except for the areas close to the mountain front. Additionally, the total rainfall data for monsoon shows a slightly decreasing trend (Figure 7c). Further, most parts of the Ramganga basin fall in the region of major groundwater loss, where the groundwater deviation from the baseline ranged from (−)18 to (−)20 mm/y (Figure 7b). In particular, the post-monsoon (Oct–May) groundwater level shows a sharp decline (Figure 7c) which is corroborated by an earlier study which showed a significant fall in both pre- and post-monsoon groundwater levels during 1999–2010 due to overexploitation [56].

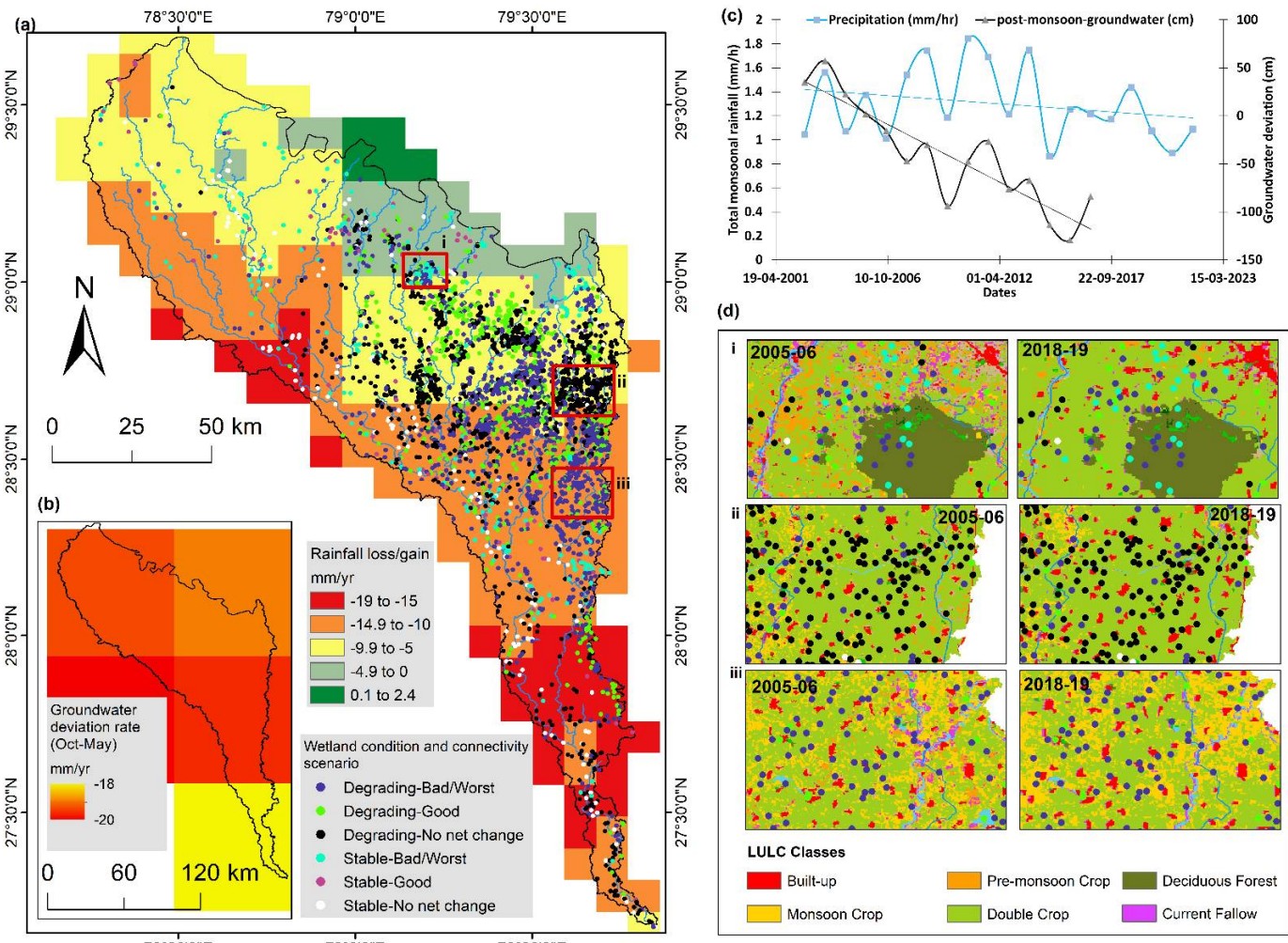

**Figure 7.** Global precipitation measurement (GPM)-derived monsoonal rainfall trends in the Ramganga plains and wetland-connectivity associations. The red boxes i, ii, and iii are the sub-areas for which detailed LULC data was analysed. (**b**) GRACE satellite-based groundwater deviation rates for post-monsoon season Oct–May. (**c**) Precipitation and groundwater trends for whole Ramganga plains—rainfall is total sum for the monsoon months (Jun–Sep) and groundwater is total sum of the post-monsoon months (Oct–May). (**d**) Some insets (**i–iii**) from (**a**) displaying the wetland-connectivity associations and LULC in two ends of the LULC time series 2005–2006 and 2018–2019.

To highlight the control of LULC changes, we have extracted the LULC data for three small windows (i, ii, and iii) for two periods, 2005–2006 and 2018–2019 (Figure 7d). A quick comparison of the maps clearly shows that there is a significant increase in both pre-monsoon and monsoon crops in all windows particularly in the region close to the

stream network. This suggests a strong linkage between cropping patterns and groundwater depletion, which is in turn related to wetland degradation (discussed later). It is therefore critical that such feedback is understood properly to design long-term strategies for wetland restoration.

## 5. Discussion

### 5.1. Controls of Wetland Degradation: A Process–Response Framework

Our connectivity analysis shows that 40.7% of the basin displays a decreasing trend in the upslope component, mainly in the northern part of the Ramganga basin. Furthermore, 56.9% of the basin area shows an increasing trend in the downslope component, and they are generally located close to the streams. A total of 17.4% of the basin area lies in the worst, 23.7% in the bad, and 20% in the good connectivity scenarios. We find two contrasting relationships between the patterns of hydrological connectivity (connectivity scenarios) and wetland distribution in the Ramganga basin. First, the regions of worst or bad connectivity scenarios match with the presence of diminishing or lost wetlands and vice versa; this demonstrates the direct control of hydrological connectivity on wetland degradation. This direct relation between surface connectivity loss and wetland degradation has also been observed elsewhere in Ganga Basin, e.g., Kaabar Tal [23,27,43], and in the lower Ganga delta region [57], in floodplains of the USA [58], China [59], and Australia [60].

In other regions of the study area, however, the inverse relationship exists, i.e., the regions of good connectivity scenarios show degrading wetlands and vice versa. This suggests the influence of local factors on wetland degradation, e.g., vertical connectivity, water abstraction or other anthropogenic factors. To explore these controls further, we use the trends of hydrometeorological data and LULC changes. While these datasets are of coarse resolution, the generalised basin-scale trends suggest that anthropogenically induced LULC changes, particularly in terms of cropping pattern, are quite significant in several parts of the Ramganga basin and have influenced wetland degradation in a substantial way. Not only have the LULC changes influenced surface hydrologic connectivity, but the increases in area under agriculture also drive groundwater exploitation. For example, in Rampur, which is one of the largest districts in the Ramganga Basin, 99.8% of the total crop area is intensively irrigated, with groundwater irrigation accounting for 97% of the total irrigation source [61]. Significant groundwater loss reduces the vertical connectivity of wetlands with groundwater system influencing the non-monsoon river inflows and hence the surface hydrological connectivity. In addition, the monsoonal rainfall has also been decreasing over time, influencing the groundwater recharge. These factors, however, need a more detailed investigation through closely spaced groundwater monitoring wells and high-resolution mapping of changes in cropping patterns.

We argue, therefore, that a wetland's hydrological health is primarily influenced by an active interaction of (a) surface connectivity (controlled by land-cover changes) and (b) sub-surface connectivity (controlled by groundwater fluctuations driven by overexploitation for agricultural use). These two factors in turn influence several other hydrogeomorphic factors through a complex process response system (Figure 8). We have grouped the factors into surface and sub-surface processes, and show their interrelationships. Among the surface processes, the LULC changes in the floodplain exert an important control in multiple ways and create a critical feedback system. In populous regions such as India, a major transformation in LULC is manifested as an increase in agriculture areas driven mainly by groundwater abstraction. Rainfall exerts the positive feedback on both upslope and downslope connectivity but has the opposite impact on wetland degradation as discussed above. Among the sub-surface controls, groundwater pumping is the most important one. The excessive abstraction of groundwater for agriculture use (driven by LULC changes) and increased demand for other purposes lowers the groundwater level, and this breaks down the vertical connectivity of the floodplain wetlands with the groundwater system, leading to their degradation. In addition, the decline in groundwater level also reduces the non-monsoonal flows in rivers, which in turn decreases stream

density. This further reduces the river–wetland connectivity which generates negative feedback leading to wetland degradation. We emphasize that it is important to understand the complex relationships among these factors to plan the mitigation measures for wetland restoration in floodplain settings.

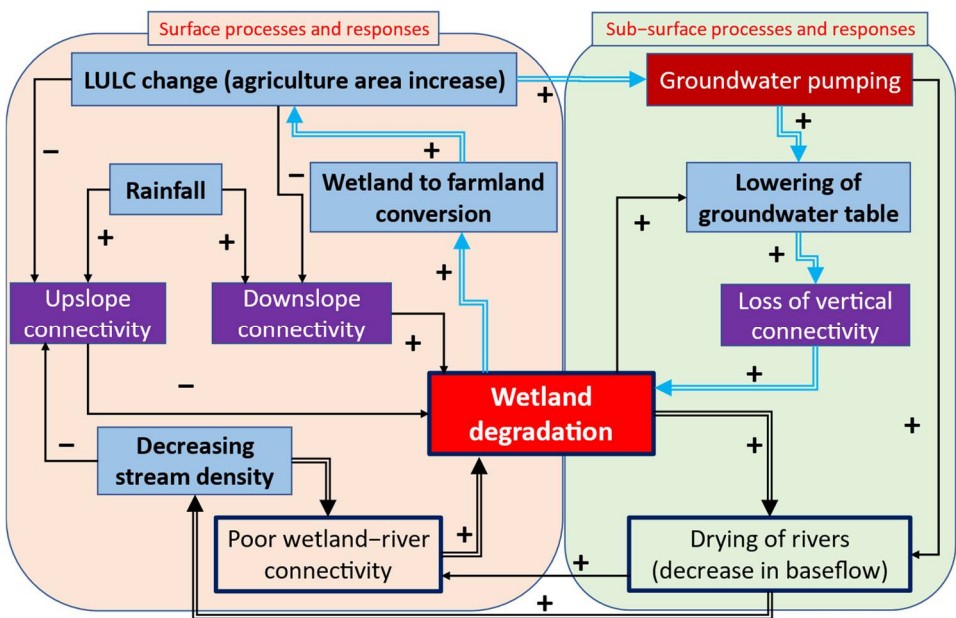

**Figure 8.** Interdependency of wetland degradation factors. Note the positive feedback among several factors responsible for wetland degradation represented by double-lined arrows. Two colours of the double-lined arrows represent two different positive feedback loops that are operating.

### 5.2. Implications for Wetland Restoration and Broader Management Perspectives

Based on this study involving the integration of wetland status in the Ramganga basin with connectivity analysis, several important management strategies for wetland restoration in floodplain settings can be formulated. The connectivity structure of the wetlands has emerged as the most crucial factor that influences the physical status of the floodplain wetlands. Therefore, a detailed analysis of floodplain–wetland connectivity and its temporal changes should be an integral component in guiding the development of wetland restoration plans. Two most important points that emerge from this work are: (a) a reduced surface hydrological connectivity of wetlands with the surrounding floodplains drives their degradation, and (b) sub-surface controls such as groundwater dynamics (vertical connectivity) also play an important role, particularly where surface hydrological connectivity alone fails to explain the wetland dynamics. Further, surface hydrological connectivity of wetlands is governed by upslope and downslope components, and the most critical control for this comes from LULC changes. Therefore, surface hydrological connectivity analysis must be accompanied by LULC change detection to identify the hotpots and buffers/barriers of connectivity. Further, since vertical connectivity of wetlands with groundwater system also plays a vital role in their hydrological status [62], a detailed investigation of groundwater level changes in response to abstraction must be carried out to pinpoint the causal factors for degradation of wetlands.

In alluvial regions, the floodplain wetlands are generally fed by river channels through surface runoff and subsurface flows. This is clearly manifested in the control of stream density and distance from the stream on wetland degradation. Therefore, river dynamics around the wetlands influences the wetlands' inflow in a significant way. It is strongly recommended that the wetland–river connectivity analysis should incorporate river dynamics studies.

Our work clearly shows that the prioritisation of wetlands for restoration should be based on multi-criteria decision-making. Apart from the physical attributes of the wetlands,

the additional factors that should be considered may include connectivity scenarios, stream, density/distance from the river and possibly the size of the wetland. We have proposed a qualitative approach for developing a district-wise inventory of the status of wetlands and the priority list for their restoration based on our previous work and conceptual understanding of wetland processes in different hydrogeomorphic settings. However, a more robust methodology involving quantitative analysis based on long-term data analysis for the factors involved and modelling the impact of possible mitigation strategies might be advantageous for wetland managers.

Large wetlands generally show significant spatial variation in degradation, and therefore, further analysis of wetland dynamics and fragmentation is necessary for developing their restoration plans. This has been amply demonstrated in some of the recent studies on large wetlands in the Ganga plains, e.g., Kaabar Tal [23] and Haiderpur [25], which have provided detailed protocols for such assessments involving multiple morphometric indices. Such analysis should primarily be aimed at identifying the spatial variability and temporal trends of wetland degradation, which could help in identifying the major causal factors for their degradation and directing their restoration efforts.

## 6. Conclusions

We have demonstrated that wetland degradation in floodplain settings is strongly influenced by its hydrological connectivity with the surrounding landscape along with hydrometeorological parameters such as long-term precipitation and groundwater dynamics. We have also developed a protocol for prioritising the wetlands for restoration and monitoring by integrating connectivity analysis with other landscape attributes such as stream density. A few important conclusions and recommendations emerging from this work are as follows:

1.  Two components of floodplain–wetland connectivity, the upslope and downslope components, influence wetland health in opposite ways. While higher upslope connectivity maintains the necessary hydrological flows, the lower downslope connectivity reduces the losses and enhances hydrological sustenance. Therefore, a sound wetland management strategy must maintain a balance between these two components of connectivity.
2.  In general, surface hydrological connectivity scenarios relate positively to wetland health, but we note several cases where a general correspondence between the two is not straightforward. In such cases, vertical connectivity of wetlands with groundwater systems seems to play an important role. This calls for serious interventions in terms of restoring the groundwater system in such regions which will provide positive feedback to wetland health.
3.  The LULC changes in floodplains, particularly the increase in agriculture areas, emerge as a critical element in the process–response framework as they provide important feedback to the groundwater system (through over-exploitation of groundwater) apart from influencing the surface hydrological connectivity itself. Therefore, the management of cropping practices and optimal groundwater utilisation must form important components of wetland restoration plans.
4.  Integrating connectivity scenarios and geomorphic indices provide valuable insights for the prioritisation of wetlands for restoration and monitoring. This should become an essential component for developing wetland management strategies. A time series analysis of several parameters based on measured data is not only rewarding to quantify the impacts, but such datasets are also necessary for monitoring the hydrological status of the wetlands. It is therefore imperative to invest significant resources in such data collection and in developing a sound monitoring protocol.

**Supplementary Materials:** The following supporting information can be downloaded at: https://www.mdpi.com/article/10.3390/w14213520/s1, Figures S1–S6: District wise maps of wetland status and connectivity scenarios, Table S1: District-wise distribution of wetland types based on the centroid location of the wetlands in the connectivity scenarios, Table S2: Restoration priority distribution (in order of priority) for wetlands in Budaun, Bareilly, Shahjahanpur, Moradabad, and Rampur districts in Ramganga basin; Table S3: Monitoring priority distribution (in order of priority) for wetlands in Budaun, Bareilly, Shahjahanpur, Moradabad, and Rampur districts in Ramganga basin, Table S4: Restoration priority distribution (in order of priority) for wetlands in Pilibhit, Udham Singh Nagar, Hardoi, Farrukhabad, Bijnor, and Amroha districts in Ramganga basin, Table S5: Monitoring priority distribution (in order of priority) for wetlands in Pilibhit, Udham Singh Nagar, Hardoi, Farrukhabad, Bijnor, and Amroha districts in Ramganga basin.

**Author Contributions:** Conceptualisation, R.S. and M.S.; methodology, M.S.; formal analysis, M.S.; resources, R.S.; writing—original draft preparation, M.S.; writing—review and editing, R.S. and M.S.; supervision, R.S.; project administration, R.S.; funding acquisition, R.S. All authors have read and agreed to the published version of the manuscript.

**Funding:** This research was funded by WWF-India, grant number WWF-I/ES/2020043 and the APC was waived by MDPI.

**Institutional Review Board Statement:** Not applicable.

**Informed Consent Statement:** Not applicable.

**Data Availability Statement:** Please refer to suggested Data Availability Statements in section "MDPI Research Data Policies" at https://www.mdpi.com/ethics, accessed on 1 September 2022.

**Acknowledgments:** We thank the WWF-India for initiating the project on wetland assessment in the Ramganga basin, which formed the basis of this work. MS was at the University of Potsdam supported by the Alexander von Humboldt Foundation and gratefully acknowledges the financial support. We also acknowledge and thank the anonymous reviewers for their critical assessment of the manuscript and the Guest Editors for inviting us to write this paper.

**Conflicts of Interest:** The authors declare no conflict of interest. The funders had no role in the design of the study; in the collection, analyses, or interpretation of data; in the writing of the manuscript, or in the decision to publish the results.

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
