# Peer review of "Integrating Hydrological Connectivity in a Process–Response Framework for Restoration and Monitoring Prioritisation of Floodplain Wetlands in the Ramganga Basin, India"

_water, doi:10.3390/w14213520_

Round 1

Reviewer 1 Report

The manuscript describes a study of connectivity applied to a large plain, the Ramganga Basin in north India. The manuscript is very well structured, well organized and clear. The objectives of the work are clearly described and the results are very well presented and discussed. The only section that needs to be improved is the Methods section: the authors refer to a number of previous published works without providing sufficient details on the contents. In general, each article has to be self-sustainable, i.e. the authors cannot pretend that the reader of a given manuscript is familiar with all previous works on the argument, therefore a complete description of the applied methods should be provided, in the main text or at least in the Supporting Information. I suggest the authors to thoughtfully revise the Methods section, adding all the details needed to the reader to understand the performed analyses.

Some specific suggestions are:

Line 45-46: it is not an "emergent property" but its study is an "emergent discipline" in case

Fig. 2: the font chosen for some of the boxes is too small

Lines 112-115: a descritpion is needed because the reader might not be familiar with the CRU concept.

Lines 130-140: please clearly define the "flow-accumulation area" and the "flow-accumulation path".

Line 144: please explain how the NDVI enters in the two previous equations.

Lines 146-148: the risk here is that the reader who is not familiar with previous works on this topic does not understand the applied methodology. I suggest the authors to summarize the CRU calculation method in an Appendix to this manuscript, or in Supplementary materials.

Lines 158-162: this is not clear. Please explain and quantify the method applied to determine the two classes. Each article must be self-sustainable, and the reader should not be asked to search for previous works to understand the methodology.

Lines 164-173: as for the letters, I don't think that a "unique" key for the three categories is needed. I have a suggestion: why not using the first letters of the alphabet for all categories? in this way the class AAA would be assigned to the wetlands in best conditions, and BDC to the worst. I believe it would be easier to understand, but this is just a suggestion.

Lines 202-203: please explain for the reader how the distance and the wetland size are used in the prioritization.

Fig. 4: the size of the characters used in the legends are so small that it is impossible to read

Fig. 6: please avoid including comments or conclusions in the figure caption.

Author Response

Reviewer 1:

The manuscript describes a study of connectivity applied to a large plain, the Ramganga Basin in north India. The manuscript is very well structured, well organized and clear. The objectives of the work are clearly described and the results are very well presented and discussed. The only section that needs to be improved is the Methods section: the authors refer to a number of previous published works without providing sufficient details on the contents. In general, each article has to be self-sustainable, i.e. the authors cannot pretend that the reader of a given manuscript is familiar with all previous works on the argument, therefore a complete description of the applied methods should be provided, in the main text or at least in the Supporting Information. I suggest the authors to thoughtfully revise the Methods section, adding all the details needed to the reader to understand the performed analyses.

Response: We thank the reviewer for their critical assessment of the manuscript. The manuscript has been revised as suggested.

Line 45-46: it is not an "emergent property" but its study is an "emergent discipline" in case

Response: We are talking about connectivity as a property of landscapes and as such, connectivity is an emergent property. The connectivity concept has emerged as an important discipline in geomorphology in last decade, but here we are not talking about it.

Fig. 2: the font chosen for some of the boxes is too small

Response: We have improved the font size and have also included the high-resolution version which should improve the readability.

Lines 112-115: a description is needed because the reader might not be familiar with the CRU concept.

Response: The CRU concept has been discussed briefly at the start of this section.

Lines 130-140: please clearly define the "flow-accumulation area" and the "flow-accumulation path".

Response: We have changed ‘flow-accumulation path’ to flow routing path which should clear the confusion.

Line 144: please explain how the NDVI enters in the two previous equations.

Response: The method section has been revised and now it includes the description of NDVI to weighting factor (W) conversion. This W is used in these two equations.

Lines 146-148: the risk here is that the reader who is not familiar with previous works on this topic does not understand the applied methodology. I suggest the authors to summarize the CRU calculation method in an Appendix to this manuscript, or in Supplementary materials.

Response: The method section has been expended and further details the CRU calculation method.

Lines 158-162: this is not clear. Please explain and quantify the method applied to determine the two classes. Each article must be self-sustainable, and the reader should not be asked to search for previous works to understand the methodology.

Response: The method section has been expended to include the details of CRU class generations.

Lines 164-173: as for the letters, I don't think that a "unique" key for the three categories is needed. I have a suggestion: why not using the first letters of the alphabet for all categories? in this way the class AAA would be assigned to the wetlands in best conditions, and BDC to the worst. I believe it would be easier to understand, but this is just a suggestion.

Response: Thanks for the suggestion. We also used this notion during an early stage of this work. However, we reserved the key ‘D’ for degraded wetlands and therefore, cannot use ABCD scheme. Also, keeping distinct code for different factors provide an opportunity to deduce the primary stressor of each wetland.

Lines 202-203: please explain for the reader how the distance and the wetland size are used in the prioritization.

Response: We have used distance in stream density calculation and suggested a way to include size and distance from stream to further narrow down the priority list.

Fig. 4: the size of the characters used in the legends are so small that it is impossible to read.

Response: We have improved the font size and have also included the high-resolution version which should improve the readability.

Fig. 6: please avoid including comments or conclusions in the figure caption.

Response: It is a good and well-accepted practice to provide some explanation in the figure captions, so that they become self-explanatory.

Reviewer 2 Report

Some scenarios are produced for connectivity-based wetlands in the study. The subject is very important and the study is valuable in terms of floodplain wetland protection but the novelty of the study is emphasized insufficiently. Some suggestions and comments to the authors are presented below:

1. Keywords should be ordered A to Z. One more keyword as can be added to keywords.

2. There are references in the abstract.

3. Check the tenses in the paragraphs. For example, there are present, present perfect and past tenses in a paragraph under “Abstract” …

4. There are some crucial and grammatical errors. Check them all.

5. Conclusions part should be improved. Main conclusions of the study should be explained well.

6. What is the novelty of the paper? It should be emphasized in the paper.

7. The resolution of figures are very low. If possible, they should be increased. Check the titles of figures. They should be separated from figures.

8. The performance metrics part is very important for the evaluation of application results. For this aim more metrics can be applied. Then, they can be given in a table. For this aim, the new and main papers are suggested below. They should be benefited and cited.

Moriasi, D. N., Arnold, J. G., Van Liew, M. W., Bingner, R. L., Harmel, R. D., Veith, T. L. (2007). Model evaluation guidelines for systematic quantification of accuracy in watershed simulations. Transactions of the ASABE, 50(3), 885-900.

Burgan, H.I., Aksoy, H. (2022). Daily flow duration curve model for ungauged intermittent subbasins of gauged rivers. Journal of Hydrology, 604, 127429. https://doi.org/10.1016/j.jhydrol.2021.127249

9. Very major editing of the language is needed.

10. Give new and last updated examples from literature about “floodplain wetlands”.

11. Use passive sentences. Check the sentences started by “we”.

12. The statistical properties as skewness, coefficient of variation, confidence intervals and distribution characteristics, etc. of used data as mean, min, max and median of used data should be given in a table.

Author Response

Some scenarios are produced for connectivity-based wetlands in the study. The subject is very important and the study is valuable in terms of floodplain wetland protection but the novelty of the study is emphasized insufficiently. Some suggestions and comments to the authors are presented below:

  1. Keywords should be ordered A to Z. One more keyword as can be added to keywords.

Response: Done.

  1. There are references in the abstract.

Response: Removed.

  1. Check the tenses in the paragraphs. For example, there are present, present perfect and past tenses in a paragraph under “Abstract” …

Response: Okay, we have edited the abstract accordingly.

  1. There are some crucial and grammatical errors. Check them all.

Response: The manuscript has been edited for all grammatical errors.

  1. Conclusions part should be improved. Main conclusions of the study should be explained well.

Response: We have sharpened the conclusion section and have highlighted the specific findings.

  1. What is the novelty of the paper? It should be emphasized in the paper.

Response: The novelty of the paper has been emphasised in the abstract and main text.

  1. The resolution of figures are very low. If possible, they should be increased. Check the titles of figures. They should be separated from figures.

Response: High resolution figures have been uploaded separately.

  1. The performance metrics part is very important for the evaluation of application results. For this aim more metrics can be applied. Then, they can be given in a table. For this aim, the new and main papers are suggested below. They should be benefited and cited.

 Moriasi, D. N., Arnold, J. G., Van Liew, M. W., Bingner, R. L., Harmel, R. D., Veith, T. L. (2007). Model evaluation guidelines for systematic quantification of accuracy in watershed simulations. Transactions of the ASABE, 50(3), 885-900. 

Burgan, H.I., Aksoy, H. (2022). Daily flow duration curve model for ungauged intermittent subbasins of gauged rivers. Journal of Hydrology, 604, 127429. https://doi.org/10.1016/j.jhydrol.2021.127249

Response: Thanks for the suggestion. This is an exploratory work aimed at basin-scale assessment and does not focus on individual wetland or a handful of wetlands. Therefore, a performance matrix cannot be generated at this stage. Also, the suggested references are not appropriate for this work.

  1. Very major editing of the language is needed.

Response: Done.

  1. Give new and last updated examples from literature about “floodplain wetlands”.

Response: The reference list has been updated and suitable additions have been made in the text.

  1. Use passive sentences. Check the sentences started by “we”.

Response: We have used passive sentences as much as possible and as appropriate.

  1. The statistical properties as skewness, coefficient of variation, confidence intervals and distribution characteristics, etc. of used data as mean, min, max and median of used data should be given in a table.

Response: Since the factors used to generate the priority lists are qualitative in nature, such descriptive statistics cannot be generated. Wetland distribution related statistics are already reported in an earlier publication (Singh and Sinha, 2022).

Reviewer 3 Report

Integrating hydrological connectivity in a process-response framework for health assessment of floodplain wetlands in the Ramganga basin, India

This MS has the potential to publish in the Water after revision. The authors have a sound knowledge of theoretical science.  It focuses on Integrating hydrological connectivity in a process-response framework for health assessment of floodplain wetlands in the Ramganga basin, India. There is no review of the relevant literature (Literature Review) to highlight what approaches have already been employed in the study area. The authors should think about if there is any improvement or something different compared with previous studies instead of just simply providing the research done in a new area. The authors should try to include this section. In the title health risk assessment on the paper, it is not properly addressed. It is suggested to add some survey-related health risk assessments of the study area.

Suggestion about MS

Introduction

The section needs major improvement. However, the sentences are not correctly interconnected for an easy flow. The introduction part does not give a clear explanation of the innovation and significance of the paper. This section not clearly described how your work is different from others or previous study. Please highlight the previous research gap of your study area in the light of gap build your new idea of your work.  Please modify it and write comprehensively.

It is arranged according to the following order.

a)         Key Problem

b)         Review of Research History

c)         Significance/ Scope

d)         My Progress/ Development

e)         Advantages

Fig 2 It is suggested to improve the resolution of this figure.

Fig 2 It is suggested to improve the resolution of this figure. Please elaborate procedure used to develop this figure.

Table 2. List of criteria used in the prioritisation algorithm.

On what basis you develop this table. It is suggested to elaborate this table.

Fig8. Please elaborate procedure used to develop this figure. 

Author Response

Reviewer 3:

This MS has the potential to publish in the Water after revision. The authors have a sound knowledge of theoretical science.  It focuses on Integrating hydrological connectivity in a process-response framework for health assessment of floodplain wetlands in the Ramganga basin, India. There is no review of the relevant literature (Literature Review) to highlight what approaches have already been employed in the study area. The authors should think about if there is any improvement or something different compared with previous studies instead of just simply providing the research done in a new area. The authors should try to include this section. In the title health risk assessment on the paper, it is not properly addressed. It is suggested to add some survey-related health risk assessments of the study area.

Suggestion about MS

Introduction

The section needs major improvement. However, the sentences are not correctly interconnected for an easy flow. The introduction part does not give a clear explanation of the innovation and significance of the paper. This section not clearly described how your work is different from others or previous study. Please highlight the previous research gap of your study area in the light of gap build your new idea of your work.  Please modify it and write comprehensively.

It is arranged according to the following order.

  1. a)         Key Problem
  2. b)         Review of Research History
  3. c)         Significance/ Scope
  4. d)         My Progress/ Development
  5. e)         Advantages

 Response: The Introduction section has been expanded as advised.

Fig 2 It is suggested to improve the resolution of this figure. Please elaborate procedure used to develop this figure.

Response: We have improved the font size and have also included the high-resolution version which should improve the readability. This figure is a flow chart that outlines the overall methodology adopted in this paper.

Table 2. List of criteria used in the prioritization algorithm. On what basis you develop this table. It is suggested to elaborate this table.

Response: The protocol used for prioritization has already been adequately explained in the main text. In our earlier work, we developed six classes of wetlands based on their physical attributes derived from the wetness index dataset: persistent, intensifying, intermittent, diminishing, new, and lost (Singh and Sinha, 2022a). In the present work, we combined these classes into two major groups - Degraded and Stable - to facilitate the prioritisation list as shown in Table 2. For degraded wetlands, we have prepared the restoration priority list, and for stable wetlands, we have prepared the monitoring priority list.

 Fig. 8. Please elaborate procedure used to develop this figure. 

Response: This figure is a conceptual diagram developed on our process understanding of the wetland environment based on the data presented in this paper and our previous work. We have redesigned this figure to separate the surface and sub-surface processes and their interactions which are responsible for wetland degradation.

Round 2

Reviewer 2 Report

There is a point that needs to be corrected in the article, after this correction the article should be accepted. According to comment 8 in the first round, the importance of performance metrics has mentioned. The authors replied this comment by “Thanks for the suggestion. This is an exploratory work aimed at basin-scale assessment and does not focus on individual wetland or a handful of wetlands. Therefore, a performance matrix cannot be generated at this stage. Also, the suggested references are not appropriate for this work.”, this is unacceptable. All suggested papers for performance metrics should be mentioned in the paper without adding or calculating any additional metrics. Without any performance metrics, the application results of the studies cannot be evaluated.

Author Response

As already pointed out in the previous review, this is an exploratory work aimed at identifying a few dozen suitable wetlands for their restoration based on wetland- and basin-scale factors. All these factors are calculated from remotely sensed datasets. At this stage, we have not used any field measurements to corroborate our results. Further, our method is also not based on simulated datasets but datasets representing actual physical processes. Therefore, it is not possible to apply either statistical or graphical model evaluation techniques recommended by Moriasi et al. (2007) which are aimed at simulated and real dataset comparisons. Similarly, the performance metrics discussed in Table 1 of Burgan and Aksoy (2022) are based on comparing modelled and observed datasets. Even if we consider our connectivity values as modelled datasets, we do not have any observed corresponding values, neither they can be calculated at basin-scale. Yes, for a plot-scale study, in a controlled environment and with very high-resolution dataset, it might be feasible to calculate actual connectivity values. However, this is not the aim or scope of the current study. While we agree that some performance evaluation matrices and statistics are desirable, we are unable to implement it here and regrettably, cannot cite these studies as well. However, we are thankful to the reviewer for this suggestion and would implement them in our future studies focused on smaller scales.